# Overcoming Immune Evasion in Melanoma

**DOI:** 10.3390/ijms21238984

**Published:** 2020-11-26

**Authors:** Kevinn Eddy, Suzie Chen

**Affiliations:** 1Graduate Program in Cellular and Molecular Pharmacology, School of Graduate Studies Rutgers University, Piscataway, NJ 08854, USA; ke112@gsbs.rutgers.edu; 2Susan Lehman Cullman Laboratory for Cancer Research, Rutgers University, Piscataway, NJ 08854, USA; 3Rutgers Cancer Institute of New Jersey, New Brunswick, NJ 08901, USA; 4Environmental & Occupational Health Sciences Institute, Rutgers University, Piscataway, NJ 08854, USA

**Keywords:** melanoma, melanoma immune evasion, immunotherapy, immune checkpoint blockade therapy, anti-PD-1, anti-PD-L1, anti-CTLA-4, adoptive T-cell therapy, T-VEC

## Abstract

Melanoma is the most aggressive and dangerous form of skin cancer that develops from transformed melanocytes. It is crucial to identify melanoma at its early stages, in situ, as it is “curable” at this stage. However, after metastasis, it is difficult to treat and the five-year survival is only 25%. In recent years, a better understanding of the etiology of melanoma and its progression has made it possible for the development of targeted therapeutics, such as vemurafenib and immunotherapies, to treat advanced melanomas. In this review, we focus on the molecular mechanisms that mediate melanoma development and progression, with a special focus on the immune evasion strategies utilized by melanomas, to evade host immune surveillances. The proposed mechanism of action and the roles of immunotherapeutic agents, ipilimumab, nivolumab, pembrolizumab, and atezolizumab, adoptive T- cell therapy plus T-VEC in the treatment of advanced melanoma are discussed. In this review, we implore that a better understanding of the steps that mediate melanoma onset and progression, immune evasion strategies exploited by these tumor cells, and the identification of biomarkers to predict treatment response are critical in the design of improved strategies to improve clinical outcomes for patients with this deadly disease.

## 1. Introduction to Melanoma

### 1.1. Melanocyte Biology and its Role in Melanoma Etiology

Stochastic accumulation of somatic mutations or inherited genetic defects allow normal melanocytes to transform into malignant melanoma. Melanocytes are the pigment forming cells of the skin that are of neural crest origin and are distributed across the epidermis, uvea, hair follicles, inner ear, heart, and mucosal tissue [1,2]. Melanocytes through various biochemical steps produce melanin within melanosomes, which is responsible for pigmentation and protection from harmful UV radiation [3,4,5,6,7]. Skin pigmentation is associated with genetic polymorphisms, between and within races, dictated by the amount of melanin produced by melanocytes and the size of melanosomes in the skin, rather than the number of melanocytes [4,8,9,10,11]. Races with darker skin pigmentation show higher melanin concentrations in their epidermis, making them less susceptible to melanoma, while individuals with lighter skin pigmentation who have a lower melanin content correspond to an increased risk for melanoma [4,8,12,13,14,15,16].

### 1.2. Melanoma Statistics and Risk Factors

Melanoma only accounts for about 1% of all skin cancers, but it is the most aggressive and dangerous one and accounts for 90% of all skin cancer deaths [14]. In the United States, it was estimated that in 2020, approximately 100,000 new cases of invasive melanoma will be diagnosed, with approximately 7000 deaths from this disease [14]. Women in general show better prognosis and overall survival than men, possibly due to sex hormone interactions with melanoma cells. There are reports suggesting that women have a stronger immune system, but the precise mechanism of this sex bias is under investigation [17,18,19,20,21,22,23,24]. In addition to gender, other risk factors for melanoma include fair skin, number of moles, UV exposure, age, and family history of skin cancer.

Moles or nevi are clusters of benign melanocytes in the quiescence state. As the number of nevi increases, the risk of developing melanoma also increases. It is estimated that about 30%–50% of all melanomas arise from nevi and are associated with non-chronically sun damaged (non-CSD) melanomas [2,25,26]. It was established that the number of nevi increases one’s risk for malignant melanoma, however, it does not guarantee that one will develop melanoma [2,27].

Both intermittent and chronic sun (UV) exposure, increases somatic mutation rates and leads to de novo cutaneous melanomagenesis, and also promotes pre-existing nevi to transform into melanoma [2,28,29,30,31,32,33]. UV signatures were detected in cutaneous melanoma, and consist of C > T transitions at dipyrimidine sites and CC > TT or (C/T)C > (C/T)T mutations [2,34]. UV exposure shows a “one-two punch” effect, enabling a pre-cancerous melanocyte to transform into a tumor cell. UV increases the mutational burden in the cells, while also locally and systemically suppressing the immune system. The relationship of UV exposure and immune escape is further discussed in Section 2.

Melanoma is a disease of old age and the stochastic accumulation of mutations within melanocytes either inherited or acquired, result in melanocyte transformation into melanoma. The average age of melanoma diagnosis in the United States is 65 years-old and death is 71 years-old, however, worldwide, melanoma incidences peaks at 70–80 years of age [14,33].

Familial history of skin cancer increases the likelihood of developing melanomas and is even higher if you have many atypical moles, also known as familial atypical multiple mole melanoma syndrome (FAMM) [35,36]. Familial melanoma comprises of 5%–12% of all melanomas [35]. Inherited genetic defects in *CDKN2A* and *CDK4* that deregulate cell cycle in melanocytes were linked with the development of familial melanoma. Another genetic disorder Xeroderma Pigmentosum (XP), is where patients have a reduced ability to repair DNA damage caused by UV [37,38,39,40,41,42,43,44,45].

### 1.3. Melanoma Diagnosis and Staging

In the clinical setting, a dermatologist diagnoses suspicious skin lesions using the ABCDE and the “ugly duckling” guidelines, also known as ABCDEF [46,47,48,49]. The ABCDEF criterion is as follows: Asymmetry, Border Irregularity, Color Variegation, Diameter Larger than 6 mm, Evolution of a lesion (changing in size, color, shape, or nevogenesis), and “Funny Looking”, where the “ugly duckling” nevi does not fit the common profiles of nevi found on a patient [46,47,48,49,50]. Dermoscope is used by dermatologists to identify possible malignant lesions [51,52,53]. Once a suspected lesion is identified, a biopsy is taken for pathohistological analysis to confirm or refute the initial diagnosis [53,54,55]. Accurate disease staging is essential for the correct diagnosis, prognosis, and treatment regimen provided to the patient. Both clinical and pathological data are assessed using the Tumor Thickness, Nodal Involvement, and Metastasis (TNM) system [56]. Tumor thickness in the TNM system considers both the thickness of the primary tumor(s) and the extent to which the tumor is ulcerated. Tumor thickness or Breslow measurement considers the depth of which the melanoma has invaded the skin, since a greater vertical depth is correlated with worse prognosis of melanoma and is associated with greater spread of the disease [56,57]. Furthermore, the extent of ulceration of melanomas gives insight to the spread of the disease, since it frees up the melanoma to grow horizontally or vertically [58,59,60]. Nodal involvement in the TNM criterion evaluates whether the melanoma has spread to the nearby lymph nodes [56]. The M in the TNM system addresses if the melanoma has spread to distant organs and lymph nodes [56]. The most common sites for melanoma dissemination are the lung, liver, brain, bones, and skin [61]. There are vast complexities in uniformly and accurately diagnosing human melanomas, therefore, the American Joint Committee on Cancer (AJCC) suggests categorizing the disease based on various permutations of the TNM system [56,62].

### 1.4. Melanoma Subtypes and Their Molecular Abnormalities

Based on primary tumor tissue location, melanoma can be broadly categorized into cutaneous and non-cutaneous melanoma. Cutaneous melanoma (sun exposed) makes up about 91.2% of melanoma cases, while non-cutaneous melanoma (sun-shielded) makes up less than 10% of all cases and have distinct genetic alterations between them [5,63,64]. Non-cutaneous melanoma has a worse prognosis than cutaneous melanoma, due to the delay in primary tumor diagnosis, the aggressive nature of these tumors, a high recurrence rate after treatment, and a poor overall survival [65,66]. Interestingly, for both cutaneous and non-cutaneous melanoma, post metastatic disease diagnosis show similar overall survival [65]. Epidemiology studies provided strong evidence that fair skinned individuals have a higher susceptibility to cutaneous melanoma, while darker skinned individuals have higher cases of non-cutaneous melanoma [22,67,68].

Cutaneous melanoma arises from transformed melanocytes on sun exposed skin and has the highest mutation burden (179 mutations per sun exposed tumor), compared to non-cutaneous melanoma (9 mutations per sun-shielded tumor) [34,69,70]. Non-cutaneous melanoma occurs in regions with low UV exposure, such as uvea, mucosal tissue, and acral tissue, and cutaneous melanoma occurs in regions more susceptible to damages by harmful UV radiation (Figure 1) [34,69,70]. Cutaneous melanoma can be further subdivided into chronically sun induced melanoma (CSID) and non-chronically sun induced melanoma (non-CSID) (Figure 1). CSID is associated with the head, neck, and the furthest extremities and individuals older than 55 years [2]. Non-CSID is associated with individuals 55 years or younger and is associated with the torso and proximal extremities [2]. The genetic abnormalities commonly associated with these two subtypes of cutaneous melanoma are *neurofibromin 1 (NF1*), *NRAS*, *BRAF* non-V600E mutations, or *KIT* in CSID, while non-CSID is associated with *BRAF* V600E mutations, suggesting that the non-CSID might originate from nevi (Figure 1) [2,71,72]. Four major genomic subtypes in cutaneous melanoma are *BRAF*, *RAS*, *NF1*, and triple wild-type (Figure 1) [34]. The *BRAF* subtype is characterized by the presence of *BRAF* hot-spot mutations (V600E, V600K, V600R, and K601E) and is mutually exclusive with *NRAS* hot spot mutations [34]. Additionally, non-hotspot mutations in *BRAF* occurred together with *N/H/K-RAS* hotspot mutations and *NF1* mutations [34]. Hot-spot mutations in *BRAF* and *N/H/K-RAS* show increased MAPK and PI3K/AKT signaling cascade activation [34]. *NF1* mutations are detected in 15% of melanoma and the majority of them are from older patients with a higher mutational burden [34]. More than half of the *NF1* mutations are associated with a loss of function [34]. Mutations in *NF1* also lead to the activation of the MAPK pathway [34]. The fourth subtype is triple wild-type with none of the mutations in the *BRAF*, *NRAS*, and *NF1* genes. Interestingly, other mutated genes were also found in this subtype: *GNAQ*, *GNA11*, *KIT*, *CTNNB1*, and *EZH2* [34]. This subtype lacks UV signatures, but the potential oncogenic drivers include structural genomic changes in copy number alterations and gene fusions [34]. Three out of the four genomic classifications engage in hyperactivating the MAPK pathway, which is involved in cell proliferation, supporting the notion that the MAPK pathway is a key player in melanoma development and progression (Figure 1) [34,73]. Other mutations that are commonly found in cutaneous melanoma are *TERT* promoter mutations, suggesting that enhanced telomerase activity leads to proliferative immortality [74,75].

Non-cutaneous melanoma arises from melanocytes located near the uvea, mucosal tissue, and acral regions. Mucosal melanoma is the rarest subtype of melanoma, comprising 1.3% of all melanoma cases, followed by acral melanoma at 2%–3%, and uveal melanoma at 5.2% (Figure 1) [5,63,76]. We recognize that acral melanoma could be considered as a subtype of cutaneous melanoma, however, the genomic profiles and risk factors/etiologies mirror other non-cutaneous melanoma subtypes, thus, we categorized it as a non-cutaneous melanoma in this review. Uveal melanoma originates from melanocytes located in the three segments of the eye—iris, choroid, and ciliary body, and is associated with older individuals, people who are fair skinned, welders, people who have light colored eyes, and is common amongst men [77,78,79,80]. Genomic abnormalities driving uveal melanoma are characterized by a low mutational burden (approximately 1.1 somatic mutations per Mb) and chromosomal gains/losses [70,81,82]. Common mutations found in uveal melanoma include *CYSLTR2*, *PLCB4*, and *GNAQ/GNA11*; all promote activation of the MAPK and PI3K/AKT signaling cascades (Figure 1) [83,84,85,86,87]. Activating mutations in *GNAQ/GNA11* also lead to activation of several transcriptional factors associated with RNA splicing, DNA damage response, and cellular proliferation [88,89].

Acral melanoma usually occurs in older darker skinned individuals and frequently have a poorer prognosis than other subtypes of melanoma, possibly due to a delay in diagnosis [76,90,91]. Acral melanoma etiology, is associated with long-term trauma/physical stress/pressure in acral regions, as well as UV exposure, which might contribute to melanocyte transformation in these regions [69,76,92]. Acral melanoma is defined as a melanoma originating from non-hair bearing skin (glabrous tissue), such as the palms, soles, or under the finger and toe nails [90,93]. This subtype is characterized by gene amplification/losses and show a low mutational burden [64,69,82,90,94,95]. Mutated genes associated with this subtype are: *KIT*, *PDGFRA*, *BRAF*, *NRAS*, *NF1*, *GNAQ*, and the *TERT* promoter (Figure 1) [82,94,95,96,97]. Amplification or deletion of many genes is a common carcinogenic process involved in acral melanoma. Genetic alterations commonly found within acral melanoma are correlated with the signaling pathways associated with cell cycle progression and cell growth. Interestingly, like cutaneous melanoma, mutations in the *TERT* promoter and *TERT* amplification might upregulate telomerase activity in acral melanoma cells, allowing them to become replicative immortal (Figure 1) [69,75,95,96].

The rarest melanoma subtype is mucosal melanoma, which originates from melanocytes located in the mucous membranes of the gastrointestinal, genitourinary, and respiratory tracts [98]. Mucosal melanoma is most commonly found amongst older women and fair skinned individuals [80,98,99]. Similar to other non-cutaneous melanoma subtypes, mucosal melanoma is characterized by a low mutational burden (2.64 mutations per Mb compared to cutaneous melanoma with 49.17 mutations per Mb), high copy number variations, and increased chromosomal structural variations [64,69,70,100]. Several genes were identified that are commonly amplified in this subtype, including *KIT*, *CCND1*, and *CDK4* (Figure 1) [64,69,97,101]. *CDKN2A* loss is often associated with mucosal melanoma [64,101]. Genes that are frequently mutated in mucosal melanoma are: *BRAF* (10%–17%), *NRAS* (5%–10%), *SF3B1*, *NF1*, *KIT* (activating mutations), *GNAQ*, *GNA11*, *TPR*, *TTN*, and *PTEN* (Figure 1) [69,101,102,103,104,105]. Additionally, Furney et al. identified mutations that were not previously identified in mucosal melanoma: *MUC2*, *UBE4A*, *PTPRT*, *NRK*, *NALCN*, *MUC4*, *MAP4K4*, *LRRC7*, *LRP1B*, *FURIN*, *CNBD1*, *CDH13*, *CACNA1C*, *AHNAK*, *ABH1B*, *KIR2DL1*, *MGAM*, and *SELPLG* [103]. Similar to acral melanoma, molecular abnormalities associated with mucosal melanoma converge on phenotypic profiles associated with hyperactivation of the MAPK and PI3K/AKT pathways, resulting in cell cycle progression and anti-apoptosis signals (Figure 1) [105]. Various subtypes of melanoma have different etiologies and genomic profiles but they converge into two major signaling pathways that were shown to play key roles in cell transformation and tumorigenesis—the MAPK and PI3K/AKT pathways [64,82].

Our group uncovered yet another partaker in melanomagenesis, the aberrant expression of a normal neuronal receptor, Metabotropic Glutamate Receptor 1 (mGluR1:protein, GRM1:gene) in melanocytes [106,107,108,109,110]. In the course of constructing transgenic mice with a fragment of genomic DNA (Clone B), which demonstrated adipocyte differentiation in vitro, resulted in concomitant deletion of 70 kb of host DNA and insertion of Clone B [106,107,108]. This disruption of host genome led to ectopic mGluR1 expression in melanocytes, in one out of five founder mice [106,107]. As this founder mouse aged, elevated pigmented lesions were detected throughout the body [106,107,111]. These pigmented lesions were histologically identified as melanocytes with high mitotic index. To confirm that the aberrant mGluR1 expression in melanocytes drives the tumor phenotype, a second transgenic line was made with GRM1 cDNA, under a melanocyte-specific promoter, dopachrome tautomerase (DCT) [108]. This second transgenic line displays similar tumor onset and progression as the first one, confirming that the aberrant expression of mGluR1 in melanocytes was sufficient to promote melanocyte hyperplasia and transformation into malignant melanoma, similar to human melanoma development [108,111,112]. Our findings that mGluR1 plays a role in melanomagenesis in mice, prompted us to examine human melanoma cell lines and biopsies for mGluR1 expression. We found 23 of 25 cell lines and approximately 60% of melanoma biopsies expressed mGluR1 at both mRNA and protein levels, independent of the BRAF/NRAS genotypes [113]. mGluR1 is a G-protein coupled receptor (GPCR) activated by L-glutamate, this receptor is normally expressed in the central nervous system and is involved in memory and learning [73]. Activation of the receptor led to stimulation of downstream effectors and hyperactivation of the MAPK and PI3K pathways [73,114,115]. Taken together, results from these studies point to the importance of delineating the stepwise molecular evolution process in the transformation of normal melanocytes into metastatic melanoma.

## 2. Conventional Melanoma Therapies

### 2.1. Surgery

Regardless of the stage at which the melanoma is diagnosed, the primary tumor is excised by local wide excision surgery, if possible, to control local disease and prevent further spread of the cancer [55,116,117,118,119]. For melanoma in situ, surgery is considered to be curative. Before or during primary tumor excision, sentinel lymph node biopsy is performed to see if the cancer cells migrated to the lymph nodes or beyond [55,116,120,121]. If cancer cells are found in the local lymph nodes, the lymph nodes surrounding the tumor area is removed (lymphadenectomy) [55,116]. Some doctors recommend the use of imiquimod cream for early stage melanoma patients who are unable to undergo surgery, in order to control the local melanoma [122,123]. The imiquimod cream promotes local anti-tumor innate and adaptive immune responses, in addition to the induction of tumor apoptosis [124,125]. If surgery can be performed, radiation therapy is sometimes recommended to prevent recurrence, by killing the remaining undetectable melanoma cells (Figure 2) [116].

### 2.2. Radiation Therapy

Melanoma is historically considered a radiation-resistant tumor type, but under certain circumstances, radiation can be used to treat melanoma [126]. These circumstances include—if a patient cannot undergo surgery, palliative therapy for late stage melanoma, and radiation given at the site of lymphadenectomy [126,127]. Melanoma is radioresistant due to the robust intrinsic DNA damage repair mechanisms [127]. Radiation therapy was shown to not only kill the targeted melanomas but also induce a systemic anti-tumor immune response against metastatic lesions, a phenomenon known as the abscopal effect [128,129]. The abscopal effect was characterized in many cancers; it is the ability of localized radiation to trigger systemic anti-tumor effect (Figure 2) [130].

### 2.3. Chemotherapy

Up until 2011, there were only two FDA-approved therapies for metastatic melanoma—dacarbazine, a chemotherapeutic agent, and high dose IL-2, an immunotherapeutic agent (Figure 2) [131]. The goal of every cytotoxic chemotherapeutic agent is to inhibit actively dividing cells, by targeting cell division or inducing DNA damage. Chemotherapy is used as a palliative/salvage therapy for late stage melanoma patients with refractory, progressive, or recurrent melanoma [132,133]. To date, no combinatorial chemotherapy regimen successfully improved the response rates, or survival, compared to monotherapy [132]. Chemotherapy induces mitotic catastrophe in cancer cells and if not repaired rapidly, it can induce apoptosis or necrosis, depending on the extent of the damage [134,135,136,137].

### 2.4. Targeted Therapy

As the cost of DNA sequencing decreases, the feasibility of genomics in cancer therapeutics increases. In the last two decades, the rapid sequencing of patient tumors unveiled a myriad set of molecular targets that were used to successfully treat malignant melanoma and other cancers. Targeted therapy is the concept of developing agents specifically attacking the drivers of carcinogenesis, to improve outcomes and reduce toxicity (Figure 2). Targeted therapy can be given as a first line therapy or as an adjuvant therapy to melanoma patients, based on their melanoma genotype and stage. The components of the MAPK pathway, specifically RAF and MEK, are druggable, and melanoma patients derived therapeutic benefits with low toxicity (Figure 2) [131,138]. In 2011, the FDA approved the first in-class targeted therapy for melanoma, vemurafenib. [131,139]. Vemurafenib was specifically approved for melanomas with *BRAF* V600E-activating mutation, this agent inhibits the kinase activity that is responsible for hyperactivating the MAPK pathway [139,140,141]. In clinical trials, vemurafenib was shown to improve survival with response rates at 48%, with manageable toxicities, when compared to the standard of care, dacarbazine [139,142]. The results of the vemurafenib trial mirrored the second-in-class BRAF inhibitor, dabrafenib [143,144,145]. Dabrafenib is approved for melanoma tumors with *BRAF* V600E/K mutations [144]. Furthermore, in clinical trials, trametinib, an MEK inhibitor, showed genotypic differences in response rates. The mutated *BRAF* melanomas had a 33% response while the wild-type *BRAF* tumors only had a 10% response [146]. In 2012, the FDA approved trametinib for mutated *BRAF* melanoma, based on the improved patient survival in the trametinib arm, compared to the standard of care [147]. Evidence suggests that in patients with mutated *BRAF*, the combination of BRAF and MEK inhibitors yields greater benefit and prolongs the development of resistance [148,149,150,151]. Recently, additional new combinations of BRAF and MEK inhibitors were approved for their usage in malignant melanoma, such as the BRAF inhibitor, encorafenib, and MEK inhibitors, cobimetinib, and binimetinib [152,153,154,155]. Like all therapeutics, patients receiving BRAF and MEK inhibitors eventually develop resistance and their disease progresses, due to acquired and tumor-intrinsic resistance mechanisms [131,138].

Another targeted therapy utilized for the treatment of melanoma are KIT inhibitors (Figure 2). A small subset of melanoma patients have DNA alterations in *KIT* that manifest as point mutations and amplifications in less than 7% of cutaneous melanoma patients, and in approximately 40% of mucosal and acral melanoma patients [97,138,156]. KIT, a tyrosine kinase, when stimulated by binding of stem cell factor (SCF) or when mutated, activates the MAPK and PI3K/AKT pathway [138]. Melanoma patients benefit from the off-label use of KIT inhibitors, imatinib and nilotinib, only after stratification based on their *KIT* mutation status [138]. *KIT* amplifications do not confer sensitivity to *KIT* inhibitors [138,157,158,159]. Growing evidence suggest that depending on the melanoma subtype, *KIT* can act as an oncogene or a tumor suppressor [160,161]. It is, therefore, important to elucidate the functions of cancer associated genes under various contexts, including tumor microenvironment, cancer types, anatomical locations, and splicing variants, since the context of a cancer associated gene affects the therapeutic viability of a targeted therapy agent [160,162].

In the era of precision medicine, it is important to characterize melanoma tumors through molecular subtypes and to identify targeted therapies that are best suited for these subtypes—*BRAF*, *RAS*, *NF1*, and triple wild-type (Figure 1) [34]. As we learn more about the genomic profiles of melanomas, these molecular subtypes must be redefined and tested against current and upcoming agents targeting oncogenes and tumor suppressors. In Section 4, we discuss immunotherapeutic agents and propose that a better understanding of each melanoma subtype biology would improve the response and survival rates for patients, given immune checkpoint blockade therapy or adoptive T-cell therapy.

## 3. Mechanisms of Immune Evasion in Melanoma

Cancer cells are constantly adapting to its host defenses, by manipulating the intrinsic and extrinsic biological pathways. Hanahan and Weinberg classified these manipulations into eight biological components—sustaining proliferative signaling, evading growth suppressors, resisting cell death, enabling replicative immortality, inducing angiogenesis, activating invasion and metastasis, reprogramming of energy metabolism, and evading immune destruction [163]. Within the last few years, immune evasion by cancer cells has become a popular and valuable therapeutic target to study. In this section, we explore the various strategies that melanoma cells utilize to escape detection by the immune system (Figure 3).

### 3.1. T-Cell Dysfunction

Cancer cells use the programmed cell death protein 1 (PD-1)/programmed death ligand 1 (PD-L1) immune-checkpoint axis to their advantage. PD-1/PD-L1 axis acts as a negative regulator of immune response and protects the host against autoimmunity [164,165,166,167]. When a pathogen infects the host, T-lymphocytes are recruited to the site of infection and exert their anti-pathogenic response. Cells in the surrounding healthy tissues protect themselves from T-lymphocytes, by expressing PD-L1, which interacts with PD-1 receptor on T-lymphocytes and prevent further T-cell activation [168,169,170,171,172,173]. In cancer, the interactions between PD-1 on cytotoxic T-lymphocytes and PD-L1 on tumor cells or tumor macrophages, NK cells, dendritic cells, and various other immune cells, result in an exhausted T-cell phenotype, rendering the immune system unable to detect and eliminate tumors via epigenetic changes within T-cells (Figure 3) [168,171,174,175,176,177,178,179,180,181,182]. In addition to the PD-1/PD-L1 interactions in the tumor microenvironment, the importance of these interactions were observed in the tumor draining lymph nodes, between the PD-1 expressing T-cells and PD-L1 expressing dendritic cells, which contribute to the anergic/exhausted T-cell phenotype [183]. Program cell death protein 2 (PD-L2) is the second ligand for PD-1 and is expressed on antigen presenting cells and melanoma cells. PD-L2 has overlapping functions with PD-L1, as a negative regulatory of cytotoxic T-cell activity [173,184,185,186]. Melanoma was one of the first solid tumors where high PD-L1 expression was detected with different expression levels, depending on the melanoma subtypes (62% in cutaneous melanoma, 44% in mucosal melanoma, 31% in acral melanoma, and 10% in uveal melanoma) [187,188]. Accumulating evidences proposes that in human melanomas, PD-L2 is more abundant than PD-L1, with a greater affinity towards PD-1, suggesting differential contribution by PD-L1/PD-L2, in regulating immune response [189,190]. Furthermore, glycosylation in PD-L1 was shown to improve the half-life of PD-L1, and strengthen its engagement with PD-1, thereby improving its ability to exhaust T-cells [191,192,193,194]. It was proposed that in the periphery tissues, cytotoxic T-cells undergoing continuous exposure to tumor antigens become activated and produce interferon-γ (IFN-γ) [187,195]. IFN-γ then interacts with the IFN- γ receptor on melanoma cells, which activates the downstream signal cascade of the JAK/STAT/IRF1 axis, stimulates the transcription factors, IRF1 and MYC, to bind to the PD-L1 promoter, and PD-L2 requires the participation of transcription factors STAT3 and IRF1 [196,197,198]. In melanomas, numerous transcription factors—HIF-1, AP-1, and NF-қβ—were implicated in regulating PD-L1 expression, albeit by varying mechanisms, as a result of differing mutational landscape [194,199,200]. In T-cells, when the T-cell receptor (TCR) is activated as a result of TCR engagement with the antigen/MHC complex, it activates the MAPK and PI3K/AKT pathway and result in T-cell activation phenotypes—transcriptional activation, cytokine production, T-cell survival, and proliferation [201,202,203,204,205]. However, in the context of cancer, when PD-L1/PD-L2 interact with PD-1 on cytotoxic T-cells, it leads to SHP1/2 recruitment to the TCR and modulates numerous phosphorylation activities, resulting in defective cytolytic T-cell function and metabolism [169,171,195,204,206,207,208,209,210,211,212,213,214,215]. In addition to PD-1, there are other immune suppressive checkpoint molecules that dampen cytotoxic T- cell activity against cancer cells—NRP-1, cytotoxic T-lymphocyte associated protein -4 (CTLA-4), TIM-3, LAG 3, and VISTA [216,217,218,219,220,221,222,223,224]. Disruption between PD-1 and one or more of these molecules result in defective development of cytotoxic memory T-cells and exhaustion of CD8^+^ T cells. Results from several studies propose that for early stage melanoma, anti-PD1/anti-PD-L1 therapy should not be given, because it would impede the development of long-term immunity. The importance of sequential treatment of anti-PD-1/anti-PD-L1 therapy with cancer vaccines was also suggested as another approach to release the brakes of the exhausted immune cells.

CTLA-4 is the second most well-known immune suppressive checkpoint regulator (Figure 2 and Figure 3). Its role as a negative regulator of the immune system is supported by the development of severe autoimmune diseases in *CTLA-4* knockout mice, stemmed from unrestrained T-cell activity [217,218]. CTLA-4 expression on T-cells, exerts its immune suppressive activities by inhibiting T-cell activation by outcompeting CD28 for the ligands, CD80/CD86—a process defined as T cell anergy [225,226,227,228,229,230].

Interestingly, the role of PD-1, PD-L1, and CTLA-4 might extend beyond their canonical functions as a negative regulator of the immune system. In melanomas, it was shown that PD-1, PD-L1, and CTLA-4 signaling might be responsible for tumor intrinsic cell proliferation, survival, growth, and metastatic signals, in addition to establishing an immunosuppressive shield around the tumor cell [231,232,233,234,235,236]. In conclusion, the notion of blocking the interactions between the immune checkpoint receptor (PD-1 and CTLA-4) and its ligand (PD-L1/PD-L2 and CD80/CD86, respectively) to reinvigorate the immune system to attack cancer cells, led to the development of anit-CTLA-4, anti-PD-1, and anti-PD-L1 checkpoint inhibitors, which is discussed further in Section 4 (Figure 2).

### 3.2. Melanoma Microenvironment Contains Many Immune Suppressive Immune Cells: Regulatory T-cells, Myeloid Derived Suppressor Cells, and Tumor Associated Macrophages

#### 3.2.1. Role of Regulatory T-cells in Melanoma Immune Evasion

CD4^+^ regulatory T-cells (Tregs) actively participate in reigning in an overactive immune response from damaging the host [237]. These same protective features are used by cancer cells for immune evasion. In melanoma, Tregs increase in the peripheral blood, lymph nodes, and tumor microenvironment, which corresponds with reduced cytolytic function of anti-tumor immune cells (Figure 3) [238,239,240,241,242,243,244,245,246,247,248]. Melanoma’s recruit/induce Tregs by secreting H-ferratin and chemoattractant cytokines/chemokines, which modulates the Treg function within the tumor microenvironment [246,249,250,251,252,253,254]. Tregs have four distinct mechanisms through which they suppress the immune system and was extensively reviewed by Vuganali et al. and Shevach [237,255]. In brief, Tregs induce immune suppression by: (1) releasing immune suppressive cytokines, IL-10, IL-35, and TGF-β, which inhibit the cytotoxic activities of immune cells, (2) induce cytolysis of immune cells, (3) target dendritic cells (antigen presenting cells), and (4) metabolically disrupt the immune cell function [237,255]. To note, in a murine melanoma model, Tregs did not affect the dendritic cell function, therefore, additional studies are needed to further delineate the mechanism(s) of Treg suppression in melanoma [256].

#### 3.2.2. Role of Myeloid Derived Suppressor Cells in Melanoma Immune Evasion

Myeloid cells are a major component of the innate immune system. These cells are responsible for protecting the host against foreign invaders, by phagocytosing pathogens and eliciting inflammatory responses to recruit other immune cells [257]. Cancer cells transform the myeloid cells found in the bone marrow, into myeloid derived suppressor cells (MDSC) [258]. MDSCs are critical in cancer progression, as they support tumor cell dissemination, and inhibit T-cell function [258,259,260,261]. In melanoma, an increase in MDSCs in the peripheral blood and tumor microenvironment is associated with disease progression, reduced T-cell function, and prognostic value (Figure 3) [260,262,263,264,265,266]. MDSCs reduce cytotoxic T-cell function in the tumor microenvironment by disrupting key metabolic pathways required for proper T cell function, which eventually result in T-cell apoptosis [267,268,269,270,271,272]. Depletion of MDSCs might improve anti-melanoma immunity, since MDSCs were shown to negative correlate with survival [263].

#### 3.2.3. Tumor Associated Macrophages in Melanoma Immune Evasion

MDSCs can differentiate into tumor associated macrophages (TAMs) and oscillate between an M1- or M2-like macrophage phenotype [273,274,275,276,277] in the tumor microenvironment [275,278]. Hypoxic regions within the tumor, push TAMs towards an M2-like phenotype, while under normoxic conditions they are pushed towards an M1-like phenotype [275,276,279]. Increased infiltration of TAMs is found within the melanoma microenvironment, as the disease progresses, specifically in M2-like TAMs, the ratio of M1/M2 TAMs are a valuable prognostic marker (Figure 3) [273,280,281,282,283,284,285]. M1 TAMs are associated with anti-tumor effects, while M2 TAMs support tumor progression. In melanomas, the enrichment of M1 gene signatures showed better prognosis than patients with enriched M2 gene signatures [286,287].

Blockade of M-CSF receptors on MDSCs preferentially direct TAMs into an M1 phenotype and GM-CSF signaling is responsible for reinforcing this phenotype [288]. Similar to the blockade of the M-CSF receptor on MDSCs, Georgoudaki et al. demonstrated that preferentially blocking the MARCO receptor by an antibody, could promote TAMs to differentiate into an M1 phenotype [289]. The MARCO receptor, a pattern recognition scavenger receptor, was associated with a gene expression profile resembling an M2-like TAM phenotype [289]. By developing an antibody against MARCO, this group was able to drive M2 TAMs into an M1 phenotype in the experimental models of melanoma and breast carcinoma [289]. Results from several studies suggest that M-CSF and MARCO receptors could regulate the PI3K/AKT/mTOR axis and polarization of M1 or M2 TAMs, however, further studies are necessary to elucidate the precise mechanisms [288,289,290,291,292,293,294].

Helper T cells (Th) regulate adaptive immune response by activating cytotoxic T-cells and phagocytic/digestive properties of the macrophages [295]. Two subclasses of the helper T cells, Th1 and Th2 are responsible for M1 and M2 polarization, respectively [286,296]. In healthy individuals and patients with surgically resected melanomas, there is a Th1 bias [297]. In melanoma patients, the Th2 subclass is predominant and leads to systemic chronic inflammation and support melanoma progression, attributed to Th2′s ability to polarize TAMs into an M2 phenotype [297,298,299]. M1 TAMs exert their anti-tumor properties by releasing proinflammatory cytokines, ROS, NO, and act as efficient antigen presenting cells, to support adaptive anti-tumor immune responses [300,301]. M2 macrophages support melanoma growth by strengthening tumor angiogenesis, inducing the Treg function to reduce cytolytic T-cell activities, and express soluble factors to dampen anti-tumor immune response [273,302,303]. As melanoma progresses into advanced stages, M1 TAMs shift to M2 TAM phenotype, to support tumor growth and tumor immune evasion (Figure 3). Developing therapeutics that can mediate the switch of M2 TAMs back to M1 TAMs might be a valuable tool to add to the arsenal of immunotherapeutic agents to improve melanoma treatment outcomes [286,289].

### 3.3. Defective Immune Recognition of Melanomas by the Immune System

Melanomas establish a process of immune editing by selecting subclones based on their capability of evading immune detection—elimination, equilibrium, and escape phases [304,305]. During melanoma progression, there is a progressive loss of antigen presentation capacity to cytotoxic T-cells by the dendritic cells, thus, reducing their immunogenicity (Figure 3) [304]. During the first phase, surveying professional antigen presenting cells, dendritic cells, detect immunogenic melanoma clones and capture these melanoma antigens. These cells then process the melanoma antigens and place them onto their Major Histocompatibility Complex II (MHC II), so they can present these peptides to naïve T-cells in the lymph nodes, resulting in the activation/expansion of melanoma-specific cytotoxic CD8^+^ T cells [304]. The equilibrium phase is when the immune system eliminates highly immunogenic melanoma clones, however, there are clones that escape anti-tumor immune responses [304]. Establishment of low immunogenic melanoma clones enables melanomas to rapidly proliferate and disseminate, a phase known as the escape phase [304].

During melanoma progression, there are various soluble factors released by tumor and immunosuppressive immune cells, which disrupt the proper function of dendritic cells in priming naïve T-cells into effector CD8^+^ T-cells in the lymph nodes (Figure 3) [256,304,306,307]. The immunosuppressive cytokine, IL-10, released by the regulatory T-cells and tumor cells, can lead to a defective antigen presentation capacity of dendritic cells (or macrophages), corresponding to reduced T-cell activation [308,309,310]. Defective antigen presentation by dendritic cells occurs as a result of the downregulation of cell surface expression of MHC II and the co-stimulatory molecules, CD80/CD86 on dendritic cells, which are essential molecules required for T cell activation [308,309]. Evidence suggest that expression of immune checkpoint molecules—CTLA-4, PD-1, PD-L1, and PD-L2 on dendritic cells, disrupts the innate immune functions and affects T-cell activation [305,311,312,313,314,315,316]. During genetic/epigenetic changes or immune editing, melanoma subclones can successfully downregulate key components of their MHC I antigen presentation pathways, and effectively escape immune surveillance (Figure 3) [317,318,319,320,321,322,323]. In concordance with this, it was observed that non-mutated melanoma associated antigens, MART-1/Melan-A, gp100, and tyrosinase, are heterogeneously expressed across melanoma cells [324,325,326,327]. During the melanoma immune editing phase, once a tumor is recognized by the immune system, it is eliminated and the subclones that have successfully downregulated MHC I or expression of another melanoma antigen, can lead to an immune refractory tumor (Figure 3). Interestingly, it was shown that melanoma cells express MHC II on the cell surface, which allows them to attract tumor-specific CD4^+^ T-cells [328]. These CD4^+^ T cells are shown to suppress the anti-tumor cytotoxic T-cell activity, by counteracting IFN-γ mediated immune response [328].

### 3.4. Spontaneous Melanoma-Prone Mouse Model Mimics Immune Dysfunction in Humans

Our lab developed various spontaneous melanoma-prone mouse models, driven by aberrant mGluR1 expression in melanocytes, which mimic melanoma development and progression in humans [107,108,329,330,331]. Two independent groups showed that these melanoma-prone transgenic mouse models accurately depict the immunological profiles of human melanoma patients [248,262,332]. Stoitzeners et al. demonstrated that an increase in immunosuppressive MDSCs within the tumor microenvironment is associated with the presence of anergic gp100-melanoma specific cytotoxic CD8^+^ T cells, consistent with melanoma patient data [262]. Further studies by this group showed an inverse relationship between the levels of DCs and the tumor burden in these mice [332]. They went on to show that rescuing DC populations within these mice by Flt3L, an endogenous small molecule that functions as a cytokine and growth factor, was able to restore cytotoxic cytokine production by T-cells [332]. Another study performed by Schrama’s group showed that as melanoma progressed in these melanoma-prone mice, there was an increase in Tregs, a decrease in CD8^+^ T cells within tumor tissues, and an increase in immunosuppressive cytokines, IL-10 and TGF-β [248]. Furthermore, as tumor burden increases in these mice, there is a decrease in CD8^+^ T-cell activation markers and lymphocyte proliferative capacity [248]. Taken together, the immune profiling data of these melanoma-prone mice, point to the notion that these mice accurately mimic the dysfunctional immune system in melanoma patients and is a good model to predict the treatment response to immunotherapy.

### 3.5. Ultraviolet (UV) Radiation-Induced Immune Suppression in Melanomagenesis

UV radiation is considered to be one of the biggest risk factors in the development of cutaneous melanoma. UV-induced accumulation of stochastic mutations in melanocytes leads to cell transformation and tumor formation. UV radiation has the capacity to induce local and systemic antigen-specific immune responses, thereby, enabling transformed melanocytes to escape immune surveillance (Figure 3) [333]. Kripke and Fisher demonstrated immune tolerance of highly antigenic UV-induced murine tumors, when allografted into syngeneic mice plus UV treatment. In contrast, in the absence of UV, the inoculated tumors were rejected [334,335,336,337]. Similar observations were made in immune-suppressed mice, suggesting that UV exposure mediates immune suppression [334,335,336,337]. Several mechanisms mediating UV-induced immune suppression were proposed, including defective antigen presentation, release of immunosuppressive cytokines, and apoptosis of immune cells [338]. UV exposure of the skin is associated with a reduction of the Langerhan cells, a type of dermal dendritic cell, at the site of exposure [333,336]. UV-exposed Langerhan cells migrate to the lymph nodes, where they are unable to activate Th1 cells, which is important in mounting an effective immune response. Instead, the Th2 cells are stimulated, which instill immune suppression by activating the regulatory T cells [338,339,340]. In addition, UV-exposed Langerhan cells are defective in antigen presentation in the lymph nodes, and undergo apoptosis when exposed to higher doses of UV, suggesting that UV exposure has the capacity to reduce the tumor antigen presentation to the immune system [333]. In addition to the modulation of immune cells into an immunosuppressive phenotype, it also showed an increase in immunosuppressive cytokines, IL-10, IL-4, and TNF-α, which were present both locally and systemically [338,340,341,342]. Furthermore, UV exposure reduced the cytokine IL-12, leading to an imbalance of Th1 and Th2 cells, with an increase in the latter [343,344]. These results propose that UV-induced activation of Th2 cells might contribute to the increased M2-like TAMs in melanoma patients. Taken together, these results suggest that when a patient receives immunotherapeutic agents, an agent that rejuvenates the immune system, the amount of sun exposure should be limited, since high UV exposure during the treatment regimen might render immunotherapeutic agent less effective.

### 3.6. Exosomes

Exosomes are nano-sized vesicles ranging from 30–120 nm, with cargo representing the cell membrane and cytoplasm (DNA, RNA, proteins, and lipids) components of the cell they originate from [5]. These are released by both normal and cancer cells, but cancer cells release greater amounts of exosomes, compared to their normal counterparts [5]. The cargo found within tumor exosomes are responsible for priming the pre-metastatic niche and suppressing anti-tumor immune response, thereby, enabling cancer cell metastasis [5]. In this subsection, we focus on the several mechanisms through which melanoma exosomes suppress anti-tumor immune responses, both locally and systemically (Figure 3).

Melanoma-derived exosomes were shown to migrate to the lymph nodes and induce tumor tolerance to prepare the lymph nodes for arrival of the melanoma cells [345,346]. Within the lymph nodes, exosomes can modulate antigen presentation, inhibit antigen-specific immune response, and upregulate immunosuppressive cytokines [345]. Melanoma exosomes were shown to transfer melanoma-derived MHC I into antigen presenting cells, downregulate the co-stimulatory molecules CD80/CD86, and upregulate immunosuppressive cytokines IL-6 and TGF-β, resulting in the defective function of antigen presenting cells, correlating with reduced T-cell proliferation [347]. Furthermore, tumor-derived exosomes, through cell surface interactions with T-cells, inhibit cell activation and induce apoptosis [348,349,350]. In addition to modulating the function of T-cells, tumor-derived exosomes reduce the cytolytic function of natural killer cells (NK) [350,351]. Tumor-derived exosomes carrying TGF-β and prostaglandin E2 induce MDSC formation, resulting in MDSC accumulation within the tumor microenvironment, and a suppressed immune response [352]. MiR-125b-5p detected in the melanoma exosome cargo can induce TAM formation, to support melanoma growth [353]. Furthermore, melanoma exosomes can induce a mixed population of M1- and M2-like TAMs, and the tumor microenvironment is responsible for reinforcing the M2-like TAM phenotype [274,276,279,354]. Melanoma and prostate cancer derived exosomes were shown to express the immune checkpoint molecule, PD-L1 on their surface, resulting in suppressed immune response, both locally and systemically [355,356].

Our lab demonstrated that inhibition of mGluR1 expression or function by genetic or pharmacological inhibitors in melanoma cells, did not modulate the number of exosomes released, but rather reduced the functions of the exosomes on the recipient cells in cell migration, invasion, and anchorage-independent growth, perhaps through cargo sorting into exosomes [357].

## 4. Understanding Melanoma Subtype Etiology and Biology to Better Treat Patients with Immunotherapies: Identification of Patient Biomarkers, Characteristics, and Combination Therapies to Improve Response Rates and Survival

Immunotherapy, a therapy that reinvigorates a patient’s own immune system to exert an anti-tumor immune response, has been around for over 130 years [358]. Early immunotherapies used biological molecules such as Coley’s toxin, IFN, and a high dose of IL-2, as well as the cancer vaccine Bacillus–Calmette–Guerin (BCG), to treat melanoma patients (Figure 2) [358,359,360]. In general, these agents were marked by low response rates but those who responded, showed a durable response [131,358,359,360,361]. Newer immunotherapies such as immune checkpoint blockade therapy and adoptive cell therapy showed remarkable anti-tumor response, corresponding to long-term durable survival, but not all patients responded to these therapies [131,172,358,362,363,364]. The inability for all patients to benefit from immunotherapy agents, suggests that better patient stratification based on patient characteristics, molecular biomarkers, and melanoma subtype, is required, to improve response rates. In light of these new immunotherapy modalities, immune checkpoint blockade therapy and adoptive cell therapy, the Response Evaluation Criteria in Solid Tumors (RECIST) guidelines that were initially developed to unbiasedly determine tumor response to cytotoxic agents, were modified to reflect the delay in adaptive anti-tumor immune response [365,366,367]. These new guidelines are known as immune-related RECIST (iRECIST) that help clinicians across multiple centers to consistently design, and manage data related to immunotherapy modalities, to ensure accurate data interpretation and analysis of the efficacy across studies [366,367]. In this section, we describe the various immunotherapeutic agents along with their known mechanism of action, molecular markers, and patient demographics, which is required to better identify patients who respond to adoptive cell therapy and immune checkpoint blockade therapy (Figure 4). Furthermore, we discuss that stratification based on melanoma subtype might improve response rates to immune checkpoint blockade therapy (anti-PD-1/anti-PD-L1/anti-CTLA-4) and adoptive cell therapy.

### 4.1. Immune Checkpoint Blockade Therapy

Antibodies that block CTLA-4, PD-1, and PD-L1 are widely used for the treatment of various cancers, including melanoma (Figure 2). These molecules are known as immune checkpoints that normally reign in an overactive immune response during infections, and reduce the likelihood of developing an autoimmune disorder [166,167,218,368]. Cancer being a chronic condition enables these immune checkpoints to be upregulated on immune cells, hence, protecting cancer cells from immune detection. Ipilimumab (yervoy), the monoclonal anti-CTLA-4 antibody, was first approved for unresectable and metastatic melanoma in 2011, soon followed by its approval as an adjuvant therapy [369]. Treatment with ipilimumab helps overcome T-cell anergy within lymph nodes and allows proper anti-tumor T-cell cytotoxicity [172]. In brief, within the lymph nodes, when CD8^+^ T-cells interact with dendritic cells, two signals are required for proper T-cell activation—TCR interaction with the MHC/peptide complex found on dendritic cells and the secondary signal of CD28 on T-cells, binding with CD80/CD86 on dendritic cells [172]. However, CTLA-4 outcompetes CD28 for CD80/CD86 binding, thereby, inhibiting the downstream TCR signaling and hampering anti-tumor CD8^+^ T cell function [226,228,370,371]. Ipilimumab binds to CTLA-4 on T-cells, which inhibits its ability to bind to CD80/CD86, allowing for the expansion of a repertoire of antigen-specific anti-tumor cytotoxic CD8^+^ T-cells and CD4^+^ T-cells, which corresponds to an improved anti-tumor immune response [372,373,374,375,376,377,378]. Interestingly, it was shown that the Fc portion of the ipilimumab antibody could deplete Tregs in the tumor microenvironment, by activating the Fcγ expressing macrophages, supporting an anti-tumor immune response [379,380,381,382]. Unfortunately, it was documented that ipilimumab treatment leads to higher immune-related adverse effects on organs, due to the promiscuous expansion of both normal and tumor cytotoxic T-cells [383,384,385]. The response rates for metastatic melanoma patients receiving ipilimumab is approximately 11%, albeit low frequency, but patients who respond display durable survival [386,387].

Monoclonal antibodies against PD-1 and PD-L1, pembrolizumab (keytruda)/nivolumab (opdivo) and atezolizumab (tecentriq), respectively, were approved for the treatment of unresectable/metastatic melanoma. Additionally, the anti-PD-1 antibodies were also approved for usage in the adjuvant setting [369,388]. The triple combination of atezolizumab, vemurafenib (BRAF V600E inhibitor), and cobimetnib (MEK inhibitor) was approved by the FDA as a first line therapy for *BRAF* V600 unresectable or metastatic melanoma [388]. It was demonstrated that the PD-1 and PD-L1 checkpoint axis acts as a negative regulator of immune response [389]. Cancer cells use this axis to their advantage to escape immune system surveillance [389,390]. Cancer cells including melanoma cells, upregulate PD-L1 expression on the cell surface. T-lymphocytes with surface expression of PD-1, interacts with PD-L1 on the tumor, leading to T-cell exhaustion, thereby, causing dysfunction of the immune system in detecting and eliminating the tumor cells [390,391]. Monoclonal antibody binding of the PD-1 or PD-L1 protein, disrupts the ligand–receptor interactions, subsequently releasing the “brakes” of quiescent cytotoxic T-cells and NK cells, to exert an anti-tumor immune response [177,392,393,394,395]. PD-1 antibodies show response rates in the range of 30%-40% which are higher than ipilimumab and showed improved overall survival [172,358,396]. Interestingly, additional studies revealed that PD-L1 blocking antibodies have a higher potency in blocking the PD-1/PD-L1 signaling than PD-1 antibodies, and pembrolizumab is more potent than nivolumab [397].

Immunotherapies, such as anti-CTLA-4 or anti-PD-1/anti-PD-L1 antibodies, have better response rates and improvement in patient survival with advanced melanoma, but clinical trial data suggest that not all melanoma patients are responsive to single agent monoclonal antibodies for CTLA-4/PD-1/PD-L1 [386,398]. Utilizing various permutations of drugs, including other immunotherapies in combination with immune checkpoint blockade therapy, might yield improvement in patient responsiveness. Furthermore, detailed characterization of melanoma patients who are sensitive to immune checkpoint therapy would improve response rates and survival outcomes for future patients, given anti-CTLA-4/anti-PD-1/anti-PD-L1 antibodies.

#### Patient Characteristics that can Improve Response Rates to Immune Checkpoint Blockade Therapy

In this subsection, we discuss patient characteristics and molecular markers that render a patient’s melanoma susceptible to immune checkpoint blockade therapy (Figure 4A). We first discuss the value of individual biomarkers in predicting treatment response, followed by the validity of testing various permutations of these markers to improve the predictive power for treatment response. The major goal of biomarker studies is to identify patient characteristics at baseline, to predict treatment response and survival outcomes [399]. Accurately identifying cancer patients who respond to therapy would improve their quality of life, reduce financial burden on non-responding patients, and most importantly, precisely identify therapies that would benefit a cancer patient who does not have the luxury of time. Meta-analysis of immune checkpoint blockade trials showed that male melanoma patients derive greater benefit than female patients from immune checkpoint blockade therapies [400]. These differences are associated with sex differences in molecular markers, such as tumor mutational burden, neoantigen load, PD-L1 expression, and density of both anti- and pro- tumor immune cells (Figure 4A) [400]. In line with this, older cancer patients and males have a better antigen presentation on their tumor cells than female and younger patients, resulting in better immune system detection and response to immune checkpoint blockade therapies [401].

Biomarkers that are associated with treatment response to ipilimumab are—densities of immune cells, molecular markers, and serum cytokine levels (Figure 4A) [399,402]. Ipilimumab treatment was shown to increase ICOS^+^ CD4^+^ T cells and enhance the ratio of effector T-cell/regulatory T-cells, correlating with clinical benefit in patients [402,403,404]. Low serum concentrations of IL-15 and TIM-3 expression on circulating T and NK cells, along with increased circulating mature NK cells, correlated with improved survival in melanoma patients treated with ipilimumab [402,405]. High baseline expression of indoleamine 2,3-dioxygenase (IDO) and FOXP3 expression in melanoma biopsies was correlated with treatment response and increase in tumor infiltrating lymphocytes [402,406]. Interestingly, CTLA-4 is not only expressed on the cell surface of T-cells but also found in the serum of metastatic melanoma patients [407]. High concentrations of soluble CTLA-4 at baseline was associated with improved overall survival but also an increased risk of immune-related adverse events [407]. Baseline levels of both anti-tumor and pro-tumor immune cell infiltrates were responsible for predicting treatment response to ipilimumab [402]. Immune-sensitive melanomas that respond to ipilimumab were marked by high baseline expression of immune associated genes and linked with pathways involved in “inflammatory response, cytotoxic T-cell mediated apoptosis of target cells, immune cell activation and migration and antigen presentation pathways” [408].

Molecular markers that are associated with monoclonal anti-PD-1 and anti-PD-L1 antibody treatment response are—PD-L1 expression, tumor mutational burden, tumor infiltrating lymphocytes, and soluble molecular factors (Figure 4A) [399,402]. Stratifying melanoma patients based on PD-L1 IHC expression on tumor cells and immune cells, is crucial in identifying responders to anti-PD-1/anti-PD-L1 therapies. Anti-PD-1-treated patients who have PD-L1 positive melanomas, have response rates of 50%–60%, and show better survival than PD-L1 negative melanomas, which have response rates of 10%–20% [179,194,409,410,411]. A similar trend was observed for atezolizumab, anti-PD-L1-antibody-treated patients [179]. However, in some patients with PD-L1-positive tumors, they do not respond to anti-PD-1 therapy, while some patients with PD-L1-negative tumors respond to therapy, suggesting that other factors also mediate the treatment responses [194,412]. PD-L1 is considered a dynamic biomarker and might not be a valuable tool to use by itself, since its expression changes can be modulated, depending on treatment, inflammation, intratumorally/intertumoral heterogeneity, temporal heterogeneity, and expression differences between the primary and metastatic melanomas [194,413]. Genomic instability is associated with treatment responses to anti-PD-1/anti-PD-L1 antibodies in melanoma (and other cancers), specifically by examining the tumor mutational burden. This concept revolves around the fact that these tumors have a higher mutation rate that increases their likelihood of presenting neoepitopes for surveying immune cells, to recognize and mount an anti-tumor immune response [414,415]. In melanoma, patients with a high tumor mutational burden responded to anti-PD-1/anti-PD-L1 therapies with improved survival, however, some patients with a high tumor mutational burden did not respond [416,417]. Panda et al. proposed that if there are specific mutations that make a tumor more responsive to immune checkpoint therapy, “immune checkpoint activating mutation threshold (iCAM)” [418]. iCAM positive tumors are defined as tumors with gene expression profiles associated with increased immune cell infiltrations, high CD8^+^ T-cells, and upregulation of immune checkpoint pathway genes. Immune cell infiltration was confirmed by histological analysis, and the iCAM-positive tumors were correlated with improved responses to immune checkpoint therapy (anti-PD-1/anti-CTLA-4) [418]. As eluded to, increased T-cell infiltration defined as a “hot tumor”, renders a melanoma susceptible to anti-PD-1/anti-PD-L1 therapy, as compared to a “cold tumor”, which has a low T-cell infiltration [419,420]. Furthermore, high eosinophil and lymphocyte counts with low lactate dehydrogenase (LDH) are associated with improved response rates and survival outcomes [402,421]. Circulating soluble PD-L1 and exosomal PD-L1 were found to be higher in melanoma patients, associated with an immune suppressed tumor microenvironment, compared to healthy donors [355,422]. Preclinical and clinical evidence propose that an imbalance of the gut microbiome with respect to presence or absence of specific microbes can modulate treatment response to immune checkpoint inhibitors [423,424,425].

Melanoma is a genetically heterogenous tumor that thrives in an environment composed of numerous cell types with various metabolic profiles, including immune cells supporting its development, growth, and progression. All these factors that participate in melanomagenesis are heterogenous, within and between patients. This might confound the use of individual biomarkers to predict treatment response. The study of biomarkers might unravel the mechanism of action or biological pathway(s) targeted by the drug(s). This helps identify why certain therapies benefit certain patients but not others. Cristescu et al. showed that a combined stratification of high tumor mutational burden plus high T-cell inflamed gene expression profiles in melanomas and head and neck squamous cell carcinoma patients, improved the identification of responders and non-responders to anti-PD-1 therapy [426,427]. Furthermore, patients within this classification also display improved progression-free survival, compared to those patients who deviated from this classification [426,427]. Similar results were observed when the patients were stratified based on the PD-L1 positivity of their tumors, high tumor mutational burden or high T-cell inflamed gene expression profiles showed improved responses and survival [426,427]. The ratio of reinvigorated exhausted T-cells in the blood to tumor burden, was proposed as an important predictor for treatment response and survival [428]. In addition to the various cellular and molecular markers used to identify responders and non-responders to immune checkpoint blockade therapy, a crucial characteristic that is not commonly addressed is melanoma subtype.

The four melanoma subtypes show distinct tumor mutational burdens, mutational profiles, PD-L1 expression, and microenvironments that impact the differential responses to immune checkpoint blockade therapies [81,188,429,430]. These subtypes can be divided into cutaneous melanoma; CSID and non-CSID with a higher mutational burden, as compared to non-cutaneous melanoma that include acral, mucosal, and uveal melanoma. In general, cutaneous melanoma shows better response rates to immune checkpoint blockade therapies than non-cutaneous melanoma (Figure 4 and Figure 5) [81,188,429,430]. Multiple factors contribute to the variance in responses to immune checkpoint blockade therapies, profiling neoepitopes and anergic/exhausted T-cells across various melanoma subtypes provide additional clues to this very complicated puzzle (Figure 5). The distinct melanoma subtype profiles suggest that cutaneous melanoma might benefit from immune checkpoint blockade therapy, while non-cutaneous melanoma benefit from adoptive cell therapy (Figure 2, Figure 4, Figure 5 and Figure 6).

### 4.2. Adoptive T Cell Therapy

The three major types of adoptive T-cell therapies are—tumor infiltrating lymphocyte therapy, engineered TCR therapy, and chimeric antigen receptor (CAR) therapy (Figure 2) [431]. These therapies subtly differ from each other, but their main goal is to enhance the cytotoxicity of cytotoxic T-cells, and other immune cells ex vivo, followed by infusion back into patients, to induce tumor regression [431,432]. In this sub-section, we discuss the different types of adoptive cell therapies for the treatment of melanoma, followed by identification of the characteristics of the patient and markers that renders a melanoma sensitive to this therapy.

#### 4.2.1. Tumor Infiltrating Lymphocyte Therapy

Tumor infiltrating lymphocyte therapy is the foundation for engineered TCR therapy and CAR therapy (Figure 2) [364]. This therapy requires the isolation of tumor infiltrating lymphocytes (TIL) from excised tumors [358,431,433]. This assumes that these lymphocytes can induce an anti-tumor immune response [358,431,433]. Isolated TILs are expanded by IL-2 treatment and then reinfused back into lympho-depleted patients, with additional treatment of IL-2 [431,434,435]. IL-2 treatment, in conjunction with TIL therapy, supports persistence of reintroduced TILs in patients, in addition to supporting cytotoxic abilities of anti-tumor CD8 ^+^ T-cells and NK cells [436]. TIL therapy in metastatic melanoma patients showed remarkable response rates ≥50%, with 22% among these patients showing complete remission [434,435]. The downsides of TIL therapy is that the tumors must be resectable, the resected tumor needs to have TILs that can be isolated and expanded, plus the overall health of the patient [358].

#### 4.2.2. Engineered TCR Therapy

To overcome the barriers of TIL isolation from tumors as well as availability of tumor-specific T-cells in patients, the engineered TCR therapy was developed (Figure 2) [358]. This therapy utilizes T-cells isolated from a patient, and genetically engineered these T-cells to express tumor-antigen-specific TCR [431]. Once modified, these cells can be expanded and infused back into the patients [432]. A major pitfall for this therapy is the reliance on the surface expression of tumor antigens on MHC I [431]. Frequently, cell surface presentation of antigens in melanoma is down-regulated and reduces its ability to induce the cytotoxic T-cell response [317,318,323]. In line with this, the response rates for engineered TCR therapy is only 13% (2/15 patients) [437]. To overcome this, CAR therapy was developed to bypass barriers imposed by tumor-surface expression of MHC I and enable T-cells to directly bind to tumor surface antigens (Figure 2) [431].

#### 4.2.3. Chimeric Antigen Receptor Therapy

The chimeric antigen receptor introduced into T-cells is engineered by the fusion of an antigen-specific recognition of monoclonal antibody domain, linked through a transmembrane domain to the components of the intracellular TCR, and the co-stimulatory molecules required for T-cell activation (Figure 2) [431,438]. This fusion receptor enables T-cells to recognize tumor antigens on the tumor surface, independent of MHC binding, enabling CAR T cells to eliminate cancer cells, regardless of MHC status [431,438]. CAR T cell therapy was shown to be remarkably successful in treating patients with B-cell malignancies, however, for solid tumors like melanoma, it was met with low response rates (19% for CARs targeting gp100 and 30% for CARs targeting DMF5), and the toxicities associated with the destruction of normal melanocytes [358,431,439]. Disadvantages in CAR T cell therapy is the time it takes to develop CAR T cells from a patient’s T-cell, cost, and toxicities [431,440].

#### 4.2.4. Patient Characteristics that can Improve Response Rates to Adoptive T cell Therapy

As we improve our understanding of adoptive T-cell therapy, the identification of prognostic biomarkers that can predict responders, non-responders, and patients who would develop resistance to therapy, has improved (Figure 4B). Patients with metastatic melanoma and have failed prior therapies, demonstrated improved survival to adoptive T cell therapy if they have a high tumor mutational burden and neoantigen load (Figure 4B) [441]. We propose that non-cutaneous melanoma, with a low tumor mutational burden, low neo-antigen load, or those that express higher levels of melanoma-associated antigens (gp100, MART1, tyrosinase, and TRP-1) would be intratumoral homogenous, rendering these tumors susceptible to elimination of antigen-specific CAR T cells (recognizes one antigen) (Figure 4B and Figure 6). This contrasts with cutaneous melanoma which has a higher neoantigen load, and is intratumoral heterogeneous, making this type of melanoma non-responsive to a single-type of antigen-specific CAR T cells (Figure 4B and Figure 6).

### 4.3. Oncolytic Viruses for the Treatment of Malignant Melanoma

Talimogene Laherparepevec (T-VEC) is the only FDA-approved oncolytic virus for the treatment of unresectable and metastatic (stage III-IV) melanoma (Figure 2). T-VEC is administered by intralesional injection directly to melanomas on the skin, or in the lymph nodes [442]. T-VEC preferentially targets, infects, and replicates within melanoma cells, without infecting healthy tissues [442,443]. T-VEC is a genetically engineered hepatitis simplex virus 1 that expresses GM-CSF [442,443]. GM-CSF facilitates the recruitment and activation of antigen-presenting cells, following T-VEC mediated lysis of melanoma cells [442,443]. Oncolytic viruses upon infection, replicate within the tumor and produce viral-associated GM-CSF, subsequently leading to tumor lysis [443]. Viral-induced tumor lysis releases tumor immunogens and GM-CSF, inducing an innate and adaptive anti-tumor immune response [443]. Preclinical studies suggest that oncolytic viruses can induce an abscopal-like effect; with tumor regression occurring at the site of injection, and induction of a systemic anti-tumor immune response that affects distant tumors [443,444]. T-VEC in the clinic showed a strong durable and objective response, with improved survival by 4.4 months, compared to the GM-CSF only treatment. Interestingly, some treated patients showed oncolytic-induced abscopal-like effects for the not-treated lesions [442,445,446,447,448]. The goal of oncovirus therapy is to convert an immunologically “cold” tumor into a “hot” tumor, or make the tumor microenvironment more immunologically active to induce a local and systemic anti-tumor immune response [443,449]. It was proposed that this therapy should be combined with different immunotherapeutic agents, cytotoxic agents, or radiation therapy, to improve clinical outcomes of metastatic melanoma patients [443,449].

## 5. Mechanistic Driven Design of Combination Therapies with Immunotherapy

In recent years, various combinatorial approaches with immunotherapeutic agents or molecular inhibitors targeting single or multiple pathway(s) entered the clinical trials for human cancers, including melanoma. Combinatorial strategies using inhibitors of non-redundant, independent pathways, without toxicity to overcome the suppressive tumor microenvironment, would be an ideal approach to improve disease treatment outcome and reduce resistance. A mechanistic-driven design of combination modalities is crucial in therapeutic treatment design, as it allows various drugs to work in concert with each other, to improve clinical response, survival, and overcome resistance. FDA approved the combination therapy of anti-CTLA-4 (ipilimumab) with anti-PD-1 (nivolumab), for the treatment of unresectable or metastatic melanoma, and resulted in improved response rates by 50%–60%, with enhanced durable survival compared to either agent alone, however, greater toxicities are detected with this combination [450,451,452,453,454,455]. The rationale for combining these two immune checkpoint blockade therapies is to expand anti-tumor cytotoxic T-cells within the lymph nodes through anti-CTLA-4 treatment, and anti-PD-1 releases the “breaks” of these effector T-cells at the tumor site to overcome the immune suppressive environment created by tumor cells (Figure 7) [450]. This combination also leads to distinct genetic and functional immune changes, as compared to anti-PD-1 or anti-CTLA-4 monotherapies [456,457].

T-VEC in combination with immune checkpoint blockade therapy has shown promising results, with improved response rates in combination therapies, compared to monotherapies in phase I and II trials [443,450]. A triple combination utilizes T-VEC plus anti-PD-1 and anti-CTLA-4, but should be staggered to reduce toxicities. A proposed treatment regimen could be, T-VEC is given initially to induce cytolysis of tumor cells releasing tumor immunogens and GM-CSF, where GM-CSF would recruit antigen presenting cells. The next treatment would be anti-CTLA-4 to induce expansion of clonal anti-tumor CD8^+^ T- cells, and finally anti-PD-1 would be administered to reduce peripheral tolerance and induce a robust anti-tumor immune response against the various clonal melanoma cell populations (Figure 7). The importance of determining the appropriate dosing schedule is essential to reduce toxicities, while also providing enough lead time to develop an immune response for the respective treatments to synergistically support each treatment modality.

Radiation therapy in combination with either anti-PD-1 or anti-CTLA-4 antibodies yielded mixed outcomes with better response in some cases, while others did not show improvement [450]. Similarly, the rational of sequential administration of T-VEC, anti-CTLA-4, and anti-PD-1, radiation therapy could be given first to promote tumor necrosis, and to induce an anti-tumor adaptive immune response. Subsequently, this could be followed by anti-CTLA-4 to induce expansion of tumor-specific cytotoxic T-cells and finally anti-PD-1 to disrupt peripheral tolerance (Figure 7). In this proposed triple combination, addition of T-VEC or GM-CSF after radiation therapy, could enhance recruitment and activation of the antigen presenting cells, therefore, strengthening the tumor-specific clonal expansion of cytotoxic T-cells [443].

Preclinical evidence suggest that adoptive T-cell transfer along with the dual treatment of anti-CTLA-4 and anti-PD-1, can improve tumor-antigen-specific cytotoxic T-cell infiltration and function within the tumor site, corresponding to the improved survival in experimental animal models (Figure 7) [458]. Adoptive T-cell transfer as a monotherapy show low response rates based on the poor infiltration of cytotoxic T-cells and function within the tumor microenvironment [458]. Albeit not directly tested, adoptively transferred T-cells along with anti-CTLA-4 treatment, mediates the expansion and improves T-cell function at the tumor site, followed by anti-PD-1 interrupts peripheral tolerance (Figure 7) [458]. Clinical evaluation of this triple combination is yet to be tested, however, in ovarian cancer, patients treated with adoptive T-cell therapy plus ipilimumab generated promising results, 1/6 patients showed partial response and 5/6 patients showed stable disease for up to 1 year [459]. Interestingly, vemurafenib, the inhibitor for mutated BRAF, paradoxically activates the MAPK pathway in adoptively transferred T-cells in a mutant BRAF-driven mouse melanoma model [460]. In this model, the inhibitor, vemurafenib, acts within its canonical function to inhibit mutant BRAF in melanoma cells but also paradoxically activates the MAPK pathway in T-cells to enhance the anti-tumor cytotoxic function of the tumor recognizing T-cells [460]. Future combination therapies can focus on how to preferentially deplete immune suppressive cells in the tumor microenvironment, to enhance the efficacy/response of immune checkpoint blockade therapy, adoptive T-cell therapy, or T-VEC therapy.

## 6. Conclusions and Future Directions

Characterizing the intrinsic and extrinsic mechanism that underlie melanoma pathology and progression, is crucial in improving the clinical outcome of patients with this deadly disease. A better understanding of melanoma biology would improve the design of novel combination therapies to improve response rate, promote tumor remission, and increase survival with a reduction in resistance development. In-depth characterization of patients who show complete, partial, or no response would unravel patient characteristics or molecular markers, which makes a melanoma (or other cancers) susceptible or unresponsive to treatments. Improved identification of responders or non-responders would improve a patient’s quality of life and ease the financial burden of cancer treatment.

Immunotherapy revolutionized the field of cancer therapeutics by its ability to induce long-term clinical response in patients who responded to treatment. Unfortunately, response rates to immunotherapy are low. However, with the ongoing biomarker studies uncovering key molecular markers such as tumor mutational burden, molecular marker expression on tumor or immune cells, circulating soluble markers, and pro- or anti-tumor immune cell populations, at baseline or on treatment, would advance the identification of responders vs. non-responders. Not surprisingly, the most common adverse event associated with immunotherapy are autoimmune disorders. Therefore, studies are being conducted to identify treatment biomarkers for immune-related adverse events, to detect and eliminate therapy-associated toxicities. In addition to molecular markers to predict treatment response, we believe that melanoma subtype should be used to stratify patients into either immune checkpoint blockade therapy or adoptive T-cell therapy (Figure 4, Figure 5 and Figure 6). Cutaneous melanoma patients including both CSID and non-CSID melanomas would benefit from immune checkpoint blockade therapy, due to it being a heterogenous tumor, with a wide repertoire of tumor-specific cytotoxic-T-cells that are restrained by immune checkpoint molecule expression (Figure 4 and Figure 5). Non-cutaneous melanoma would benefit from adoptive T-cell therapy, because of its low tumor mutation burden, and would have a higher expression of melanoma-associated antigen (Figure 4 and Figure 6). Therefore, there would be a higher response to adoptive T-cell transfer because this melanoma subtype contains a more homogenous T-cell population (Figure 6). Multiple biomarkers should be used to predict treatment response or identify immune-related adverse events. Biomarker identification would help tease out the mechanism of action of these immunotherapeutic agents. Better understanding of the mechanism of action of immune checkpoint blockade therapy, adoptive T-cell therapy, and T-VEC therapy would improve the design of combination therapies with other immunotherapy agents, targeted therapies, radiation therapy, or chemotherapy.

## Figures and Tables

**Figure 1 ijms-21-08984-f001:**
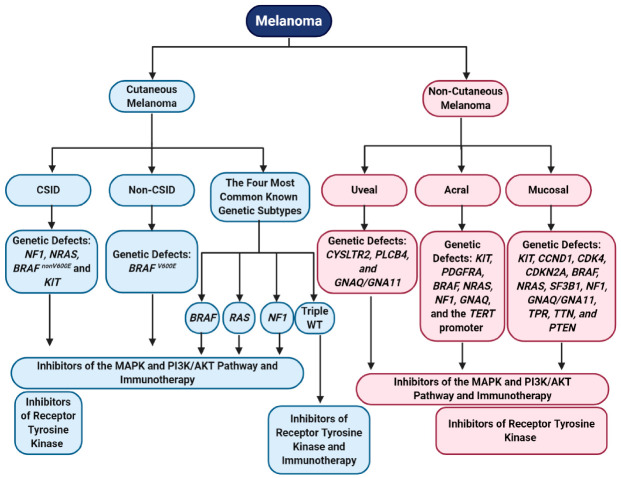
Melanoma can be segregated into distinct subtypes, based on anatomical location, sun exposure, and genetic profiles, which affect treatment responses to MAPK, PI3K/AKT, receptor tyrosine kinase inhibitors, and immunotherapies. CSID—Chronically Sun Induced melanoma; non-CSID—Non-Chronically Sun Induced melanoma; and triple WT—Triple Wild-Type. Created with BioRender.com.

**Figure 2 ijms-21-08984-f002:**
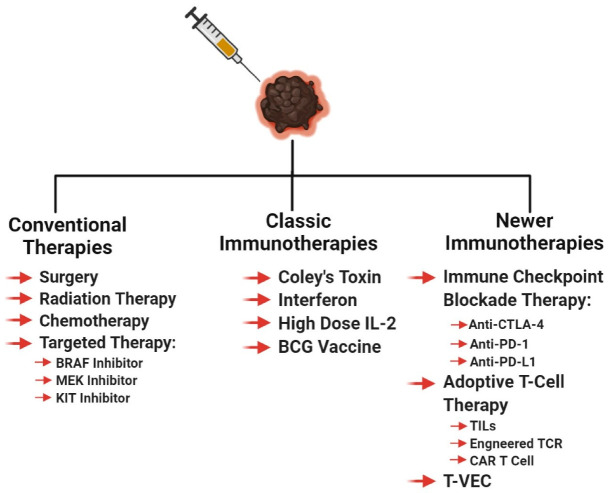
Various melanoma therapies used in the clinic. These therapies include the standard treatments along with the precursors for modern-day immunotherapies, followed by “targeted” immunotherapy. Created with BioRender.com.

**Figure 3 ijms-21-08984-f003:**
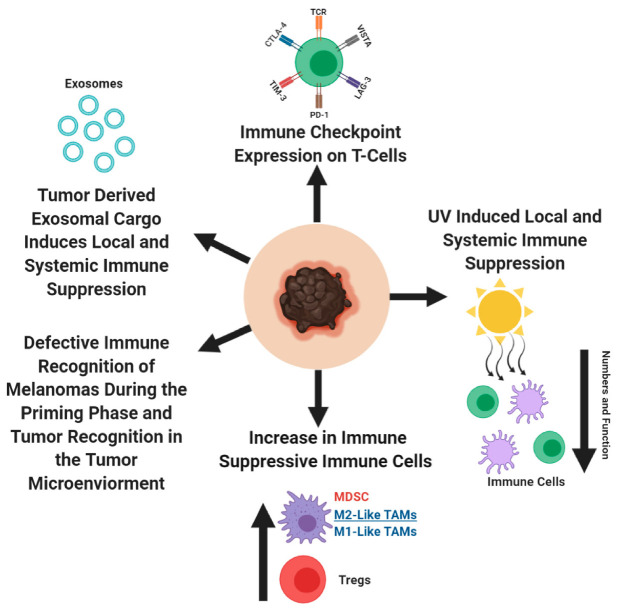
A summary of the immune dysfunctions that contribute to melanoma development and progression. Created with BioRender.com.

**Figure 4 ijms-21-08984-f004:**
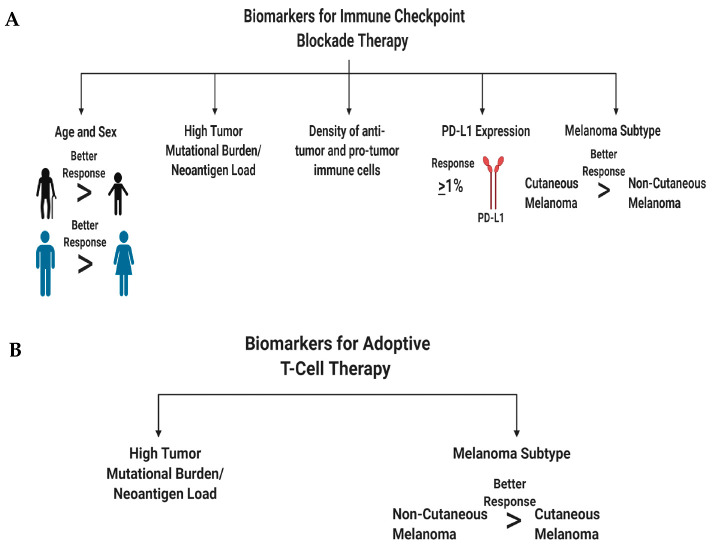
Biomarkers for immune checkpoint blockade therapy (**A**) and adoptive T-cell therapy (**B**) that can improve stratification of melanoma patients into responders and non-responders. Created with BioRender.com.

**Figure 5 ijms-21-08984-f005:**
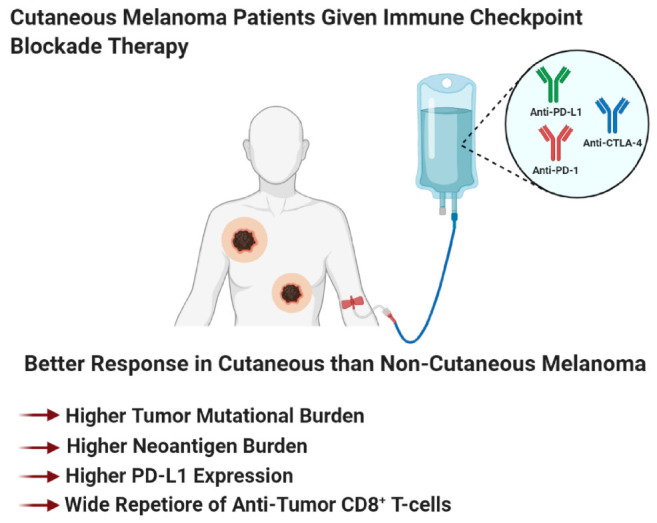
Cutaneous melanoma patients benefit from immune checkpoint blockade therapy better than non-cutaneous melanoma patients. Created with BioRender.com.

**Figure 6 ijms-21-08984-f006:**
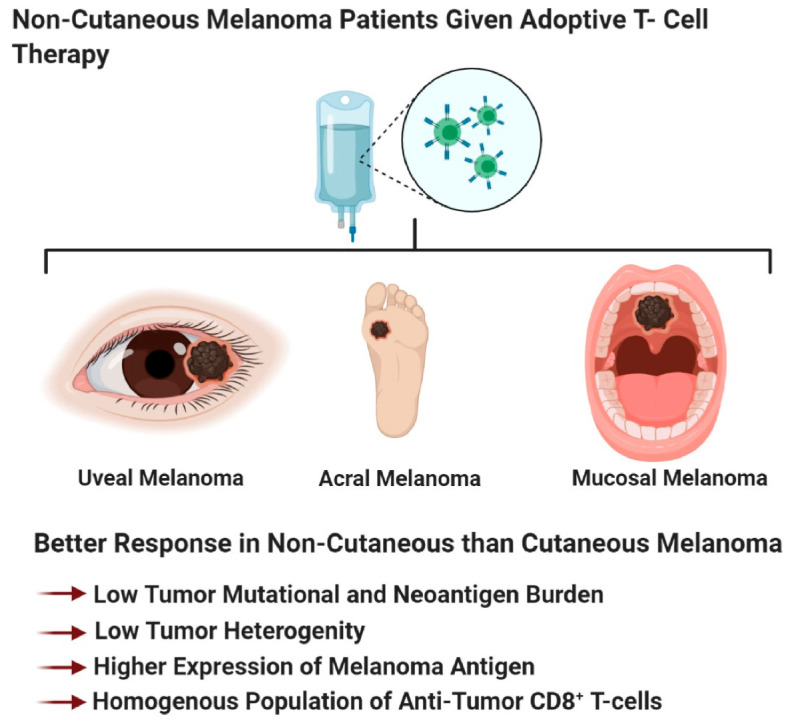
Non-cutaneous melanoma patients benefit from adoptive T-cell therapy better than cutaneous melanoma patients. Created with BioRender.com.

**Figure 7 ijms-21-08984-f007:**
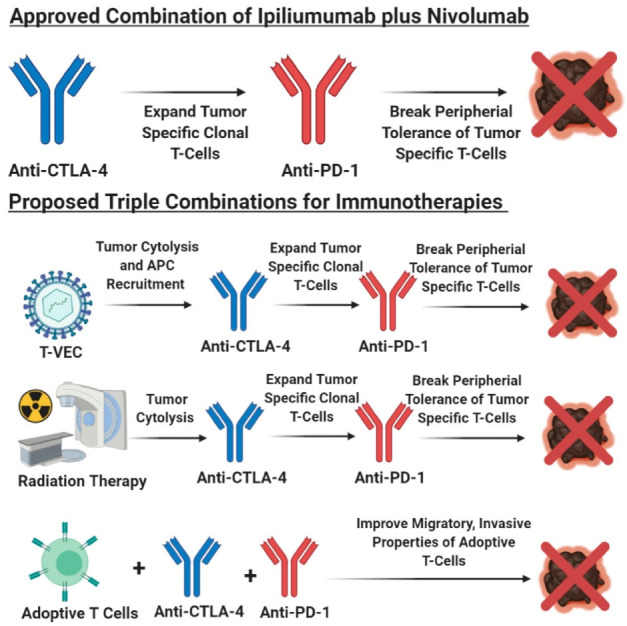
Mechanistic-driven design of combination immunotherapies. Ipilimumab plus nivolumab are approved for the treatment of metastatic melanoma, while the other three are currently under investigation or are proposed in this review. Created with BioRender.com.

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
