# Peer review of "Overcoming Immune Evasion in Melanoma"

_ijms, 2020, doi:10.3390/ijms21238984_

Round 1

Reviewer 1 Report

The authors have already published a review of melanoma progression in the journal “cancers”. In this review, the novelty is the focus on the mechanisms of immune evasion in melanoma.

To this aim, this reviewer strongly suggests reducing the introduction part as many of the subjects are mainly of general knowledge and unnecessary for the focus of this review. 

It would be useful to make a scheme of the melanoma subtype and particularly of genetic subtypes to understand which therapies can be then used according to the patient’s type of melanoma.

Also, a scheme of the possible immunotherapy used according to the melanoma subtype is necessary.

Moreover, imiquimod cream is not cited in the text, however, is FDA approved and used for early-stage melanoma treatment.

Please pay attention to the abbreviations, i.e. page 9 line 373, TCR is not introduced, while it is later at p15 line 644/645. Please check through the paper.

Fig. 2 there is a typing error “UV induced local and systemic immunr..” please correct the word “immunr”.

Reviewer 2 Report

This manuscript entitled “Overcoming Immune Evasion in Melanoma” by Kevinn Eddy and Suzie Chen review the background, immune evasion and immunotherapy in melanoma. Although no restriction of words for this journal, I considered the authors wrote too much in the review. In addition, the authors only summarized the literature without much personal opinions and suggestions which did not meet the criteria for this journal. Furthermore, the authors did not make correct statements in the review so some statement may the readers confused. The authors may not be familiar with clinical treatment so the section of treatment of melanoma is quite difficult to read. Unless the authors can correct all the misunderstandings, I don’t consider this review is good enough for publication.

Some examples for the errors

Line 19 Ipilimumab, Nivolumab, Pembrolizumab, and Atezolizumab are not shown with capital for the first letter.

Line 69: It has been established that the number of nevi increases “your” risk to malignant melanoma, however, it does not guarantee one will develop melanoma

Line 72 also increase the nevi to transform into melanoma?

Line 75, intermittent UV exposure is a player?

Line 83: Cancer is a disease of old age and stochastic accumulation of mutations within melanocytes à wrong statement

Line 92: The gene names should be presented with italic, such as CDKN2A and CDK4

Line 118 both regional and distant metastasized lymph nodes are good prognostic markers à distant is not the N stage. What is the reference

Line 120 Important to note that circulating tumor cells have been detected in melanoma patients with only regional lymph node metastasis or no metastasis at all – unclear

Line 128: distant lymph nodes ?

Line 147-150: The genetic abnormalities within the two subtypes of cutaneous melanoma are: neurofibromin 1 (NF1), NRAS, BRAF non V600E, or KIT in CSID while non-CSID is associated with BRAF V600E, suggesting that the non-CSID may originate from nevi. This statement is not correct. BRAF mutation is not limited in non-CSID.

Line 160 GNAQ, GNA11 is common mutated in uveal melanoma not cutaneous melanoma.

Line 163-165 Another mutation that is commonly found in cutaneous melanoma are TERT promoter mutations, suggesting that enhanced telomerase activity leads to proliferative immortality. I don’t consider this statement of the association is correct.

Line 170-175 “we recognize….we recognized” The statement is strange.

Line 183: UV signatures has not been associated with acral melanoma etiology. Reference ?

Line 238: this statement is not true. For unsectable or metastatic melanoma, surgery is not main treatment in these cases.

Line 241. Sentinel lymph node dissection? no such medical method. Only sentinel lymph node biopsy should be performed. In addition, this is usually done before not during primary tumor resection.

Line 252: RT is used for 4 circumstances but first three seem to be the same ?

Line 269 patients with refractory, progressive and ? refractory to which ? progressive after ?  

Line 272 “The proposed mechanism” should be moved after “division or inducing DNA damage”(Line 268)

Line 283: stages I-IV patients à is not correct

Line 284: MEK activating mutations à not correct

Line 297: than either agent alone à not correct

Line 301: within a year à not correct

Line 310: KIT inhibitors are not approved by FDA.

Line 311: approximately 40% à not so high

Line 313 Upon binding of cytokine stem cell factor (SCF) à not needed for KIT mutation

Line 318 KIT can act as an oncogene or a tumor suppressor à what dose it mean? Authors should clarify this.

Line 357 melanoma cells.?

Line 396-398: PD-1, PD-L1, and CTLA-4, why the authors use italic.

Line 682 only a small proportion of melanoma patients à not really ture

Line 700: Sex is associated with immune profiling but not a biomarker to predict response from ref. 410. In addition, how can we improve the sex to improve response ?

Line 729: anti-PD-L1: the author should specify which drugs as only one anti-PD-L1 antibody is approved. However, in the reference, they use another antibody. In addition, the authors should describe the assays used for PD-L1 expression. Different assays should different results so author should specify the assays used in the study.

Line 740: In melanoma, patient with a high tumor mutational burden have shown to respond to antiPD-1/anti-PD-L1 therapies with improved survival  à not always true

Line 778: cutaneous melanoma show à showed or shows

Reviewer 3 Report

The authors in the manuscript entitled “Overcoming Immune Evasion in Melanoma” reviewed the disease biology of melanoma and therapies to combat this disease. The authors highlighted different mechanisms by which melanoma evades the immune response and how better stratification of patients based on their disease characterizing mechanisms and melanoma subtypes would lead to better clinical outcomes of combination therapies. The manuscript comprehensively described melanoma pathobiology and approved treatment strategies. The manuscript is written very clearly. I think that this review would interest the readers of IJMS. Proofreading may be needed to shorten the lengthy sentences.

Round 2

Reviewer 2 Report

This manuscript entitled “Overcoming Immune Evasion in Melanoma” by Kevinn Eddy and Suzie Chen review the background, immune evasion and immunotherapy in melanoma. The authors have largely revised and improved. However, the authors usually replied they did not claim something. It is true if you just read the single sentence. But if you read the whole section or whole paragraph or the sentences before or after, that would confuse the readers such as me. And the most important issue is that the authors lack of clinical experience so they can only address the clinical points I raised. For examples, the authors reply that the patients with recurrent melanoma, the patients with inoperable melanoma, and the patients undergoing palliative treatment are mutually exclusive. Actually, they are almost the same. I suggest the author consult the clinician to revise this paper.

  1. The authors should recheck and correct all the Ipilimumab, Nivolumab, Pembrolizumab, and Atezolizumab in whole manuscript and they should not be shown with capital for the first letter.
  2. Cancer is a disease of old age and stochastic accumulation of mutations within melanocytes à I meant the last-half sentence: “Cancer is a disease of stochastic accumulation of mutations within melanocytes” is wrong ? It should be “Cancer is a disease of old age and melanoma is a disease………………..?
  3. The reply from author: N stands for metastatic disease in both regional lymph nodes and non-nodal locoregional sites. This is still wrong statement !! But this sentence is not shown in the manuscript.
  4. non V600E à non-V600E
  5. If the authors did not claim BRAF V600E mutation is limited to non-CSID, how do the author suggest that the non-CSID may originate from nevi.
  6. Another mutation that “is” commonly found in cutaneous melanoma “are” TERT promoter mutations, suggesting that enhanced telomerase activity leads to proliferative immortality. I agreed with authors’ reply. But the gramma may be incorrect.
  7. UV signatures has not been associated with acral melanoma etiology. After I checked the reference the authors provided, the UV signatures can also be found in acral melanoma.
  8. Regarding the surgery for stage IV patients, I don’t agree authors’ reply as the surgery is not the “mainstay” treatment. Surgery can be performed for palliative purpose.
  9. I agree with authors the change from SLN dissection to SLN biopsy. But the term “migrated” is strange here.
  10. RT is used for 4 circumstances but first three seem to be the same? I don’t agree with authors’ reply.
  11. the combination of BRAF and MEK inhibitors yielded greater benefit than ? the grammatic error.
  12. within a year à still in correct. The median PFS of dual targeted therapy is around 10 months. But this number is median !!!!! That doesn’t mean all the patients experience resistance within one year.
  13. As KIT inhibitors are not approved by FDA, I suggested the authors should mention this important point.
  14. approximately 40% à not so high à the authors still did not reply this point. KIT mutation accounts for much lower than 40%.
  15. Upon binding of cytokine stem cell factor (SCF) à not needed for KIT mutation à the sentence followed by KIT mutation may confuse the reader.
  16. KIT can act as an oncogene or a tumor suppressor. According to authors’ reply, I suppose the KIT function varies based on the different genetic alteration not melanoma subtype. I don’t consider the same KIT mutation can have distinct function in various subtype of melanoma.
  17. Melanoma cells.? à In this section, the authors discuss the immune regulation in cancer. The authors used cancer cells in whole paragraph but melanoma cells in this sentence. That is my question why the authors only emphasize melanoma cells here.
  18. Response 30: only a small proportion of melanoma patients à In the study of checkmate067, this most important clinical phase III study, the response rate for nivolumab is 43.7% and ipilimumab is 19.0%. How can the authors claim 43.7% is a small proportion.
  19. The authors did not claim sex is a biomarker, but “male melanoma patients derive greater benefit than female” mean sex is a predictive or prognostic factor. The authors keep this In the section “Patient Characteristics which can Improve Response Rates to Immune Checkpoint Blockade Therapy” but I don’t consider sex can be improved. Also, the authors keep sex in the figure 4. How can authors claim sex is not a biomarker ?

Round 3

Reviewer 2 Report

no more comments